# *Lactobacillus paracasei* A13 and High-Pressure Homogenization Stress Response

**DOI:** 10.3390/microorganisms8030439

**Published:** 2020-03-20

**Authors:** Lorenzo Siroli, Giacomo Braschi, Samantha Rossi, Davide Gottardi, Francesca Patrignani, Rosalba Lanciotti

**Affiliations:** 1Department of Agricultural and Food Sciences, University of Bologna, p.zza Goidanich 60, 47521 Cesena, Italy; lorenzo.siroli2@unibo.it (L.S.); giacomo.braschi2@unibo.it (G.B.); samantha.rossi2@unibo.it (S.R.); davide.gottardi2@unibo.it (D.G.); rosalba.lanciotti@unibo.it (R.L.); 2Interdepartmental Center for Industrial Agri-food Research, University of Bologna, Piazza Goidanich 60, 47521 Cesena, Italy

**Keywords:** *Lactobacillus paracasei*, high-pressure homogenization, stress response, membrane fatty acid composition

## Abstract

Sub-lethal high-pressure homogenization treatments applied to *Lactobacillus paracasei* A13 demonstrated to be a useful strategy to enhance technological and functional properties without detrimental effects on the viability of this strain. Modification of membrane fatty acid composition is reported to be the main regulatory mechanisms adopted by probiotic lactobacilli to counteract high-pressure stress. This work is aimed to clarify and understand the relationship between the modification of membrane fatty acid composition and the expression of genes involved in fatty acid biosynthesis in *Lactobacillus paracasei* A13, before and after the application of different sub-lethal hyperbaric treatments. Our results showed that *Lactobacillus paracasei* A13 activated a series of reactions aimed to control and stabilize membrane fluidity in response to high-pressure homogenization treatments. In fact, the production of cyclic fatty acids was counterbalanced by the unsaturation and elongation of fatty acids. The gene expression data indicate an up-regulation of the genes *accA*, *accC*, *fabD*, *fabH* and *fabZ* after high-pressure homogenization treatment at 150 and 200 MPa, and of *fabK* and *fabZ* after a treatment at 200 MPa suggesting this regulation of the genes involved in fatty acids biosynthesis as an immediate response mechanism adopted by *Lactobacillus paracasei* A13 to high-pressure homogenization treatments to balance the membrane fluidity. Although further studies should be performed to clarify the modulation of phospholipids and glycoproteins biosynthesis since they play a crucial role in the functional properties of the probiotic strains, this study represents an important step towards understanding the response mechanisms of *Lactobacillus paracasei* A13 to sub-lethal high-pressure homogenization treatments.

## 1. Introduction

High-pressure homogenization (HPH) is one of the most promising alternatives to traditional thermal treatments for food preservation and product innovation due to the physico-chemical, structural changes following the treatment. In the food sector, it has has been proposed for several purposes including microbial, enzyme and bacteriophages inactivation [1,2,3], large scale cell disruption for the recovery of microbial intracellular metabolites and enzymes [4,5] and improvement of the food safety, shelf-life, texture and functional properties [6,7,8]. Due to the phenomena of cavitation, shear stress, turbulence, and impingement that take place during the treatment, HPH result in a strong antimicrobial activity [7,9], modulation of some enzyme activities and maintenance of color, flavor, and nutritional/functional properties of the treated matrices [10,11]. The microbial inactivation caused by the application of HPH, although affected by several factors and mainly by the physio-chemical features of the food matrix and the sensitiveness of different microorganisms, increases with the pressure level [12].

HPH treatments, at levels lower than 100 MPa (sublethal pressures), applied to microbial starter, co-starter or probiotic cultures of lactic acid bacteria (LAB) and yeasts, are reported to change their metabolic and enzymatic activities resulting in the production of probiotic fermented milks or cheeses with improved sensorial, technological, or functional properties. In fact, HPH, as a versatile approach, was able to modulate the autolytic phenomena and volatile molecule production by yeast starters used to produce sparkling wines according to the traditional method, or to obtain yeast lysates for oenological application, respectively [10,13,14]. Sub-lethal HPH treatments applied to LAB demonstrated to be a useful strategy to regulate, in a strain-dependent manner, fermentation kinetics, proteolytic activities and volatile molecule profiles of starter cultures and non-starter LAB, without detrimental effects on their viability [4]. Concerning the probiotic features, sub-lethal HPH treatments improved acid and bile tolerances of *Lactobacillus acidophilus* LA-K [15] and enhanced some technological and functional properties of *Lb. acidophilus* 08, *Lb. acidophilus* DRU and *Lb. paracasei* A13 [16,17]. 

For example, HPH treatment performed at 50 MPa on *Lb. paracasei* A13 was able to increase bacterial hydrophobicity, auto-aggregation capability and tolerance to simulated upper gastrointestinal tract, in in vitro trials. Moreover, a similar treatment improved its fermentation rates in butter-milks, milk and cheese curds and its viability during refrigerated storage both in real models and real food systems. In fact, HPH-treated cells of *Lb. paracasei* A13, added during the manufacturing of caciotta cheese, maintained a high viability at refrigerated conditions, up to 14 days, and accelerated the ripening kinetic, conferring a better quality to the final product. Finally, treated bacteria were more resistant to simulated gastrointestinal digestion in vitro trials compared to untreated cells present in the same food matrix [18]. Sub-lethal HPH treated cells of *Lb. paracasei* A13 were able to modulate the murine immune system inducing a high IgA response, compared to untreated cells. In fact, modifications of the outermost cellular structures by the hyperbaric treatments play an important role in the final probiotic and immune cells interaction [15,17]. Tabanelli et al. [17] described the modification of membrane fatty acid (FA) composition and an increase of unsaturated fatty acids (UFAs) in sublethal HPH treated *Lb. acidophilus* DRU and *Lb. paracasei* A13. Moreover, changes of peptides and volatile molecules profiles, using MALDI-TOF MS (matrix-assisted laser desorption ionization time-of-flight mass spectrometry) and GC/MS-SPME analyses, were reported in probiotic lactobacilli [19]. Although literature data showed that these modulations and modifications are the main response mechanisms of probiotic lactobacilli to HPH, the gene response adopted by *Lb. paracasei* A13 can be fully understood only clarifying the relationship between gene expression and changes in the physiological *status* observed. 

In this context, the aim of this work was to clarify in *Lb. paracasei* A13 the relationship between the modification of membrane FA composition and the application of different levels of high-pressure homogenization. For this, the expression of some genes involved in the FA biosynthesis in *Lb. paracasei* A13 was studied. In fact, maintenance of membrane fluidity and functionality is fundamental to face and counteract the environmental stresses, including HPH treatment, but it is also crucial for the probiotic activity and its technological performances in food systems.

## 2. Materials and Methods 

### 2.1. Microorganisms and Culture Preparation 

*Lb. paracasei* A13, used in this experimental work, is a commercial probiotic strain isolated from dairy Argentinean dairy products. Stock cultures of this strain were maintained at −80 °C in de Man, Rogosa and Sharpe (MRS) broth (Oxoid, Basingstoke, Hampshire, Uni-ted Kingdom) with glycerol (20% *v*/*v*). Fresh cultures were prepared by two consecutive passages of a 1% (*v*/*v*) inoculum of the frozen stocks in MRS broth, with incubation at 37 °C for 18 h under aerobic conditions but without agitation. Then the strain was inoculated (1% *v*/*v*) in a final volume of 1 L of MRS broth at 37 °C for the other 18 h. 

### 2.2. High-Pressure Homogenization (HPH) Treatment 

The *Lb. paracasei* A13 cells from overnight cultures were harvested by centrifugation (8000 rpm × 15 min) and washed twice with physiological solution (NaCl 0.9% *w*/*v*). The pellet was resuspended in sterile physiological solution, to reach a final concentration of about 9.0 log CFU/mL, and then treated with HPH. The homogenizing treatments were performed using a continuous high-pressure homogenizer PANDA (Niro Soavi, Parma, Italy). The inlet temperature of the cell suspension was 25 °C with an increasing rate of 2.5 °C/10 MPa. Cells were treated at 50, 150 or 200 MPa for few milliseconds and cooled by using a thermal exchanger (Niro Soavi, Parma, Italy) resulting in an outlet temperature of 15 °C. After the treatment, the cells were further cooled down for 1 min to reach 10 °C using an iced water bath. Cultures treated at 0.1 MPa were used as control.

The cell load of *Lb. paracasei* was detected before and immediately after the HPH treatments by serial decimal dilution in physiological solution and plating on MRS agar (37 °C, 48 h, aerobic conditions).

### 2.3. Membrane Fatty Acids Analysis

Immediately after HPH, samples were collected for membrane FA analyses. Lipid extraction and membrane FA analyses were performed according to the method of Suutari et al. [20]. GC–MS analyses were carried out on an Agilent 6890 gas chromatograph system (Agilent Technologies, Palo Alto, CA, USA) coupled to an Agilent 5970 mass selective detector operating in electron impact mode (ionization voltage, 70 eV), according to Serrazanetti et al. [14]. The lipids were fractioned by solid-phase extraction (SPE). For GC analyses, a GC-Mass Agilent 6890 gas chromatograph (Agilent Technologies, Palo Alto, CA, USA) coupled to an Agilent 5970 mass selective detector operating in electron impact mode (ionization voltage 70 eV) was used. The column used was a Chemtek RTX-2330 (260 °C: 30 m × 250 µm × 0.2 µm; 10 cyanopropyl-90 biscyano: 236.56493). The injector and detector were both held at 250 °C. The temperature was programmed from 120 °C (held for 5 min) to 215 °C at a rate of 4 °C/min, then from 215 °C to 225 °C at a rate of 0.5 °C/min, and the final temperature was held for 5 min. The carrier gas was helium with a rate of 1 mL/min and a split ratio of 1:10. Compounds were identified using the National Institute of Standards and Technology (NIST11) and comparing their retention times with those of the Bacterial Acid Methyl Ester (BAME) Mix standards (Sigma-Aldrich, Milan, Italy). The concentration of each FA (ppm) was estimated using an internal standard (C13:0, 50 ppm final concentration).

### 2.4. RNA Extraction and Reverse Transcription 

Immediately after HPH, samples were collected for RNA extraction and reverse transcription. RNA was extracted from each sample, previously treated with lysozyme, using the SV Total RNA Isolation System (Promega, Fitchburg, WI, USA). The yield and the purity were determined by measuring the absorbance at 260 nm and the ratio at 260/280 nm using the BioDrop μLITE (BioDrop, Milan Italy). All the samples had a yield of around 15 ng/μL and only those with a ratio of 260/280 nm above 1.9 were used for reverse transcription. The reverse transcription of RNA into cDNA was performed according to Serrazanetti et al. [14]. Based on the final concentrations, 1 μg of RNA was used to obtain cDNA with 2 μg of random primers (Promega),40 μM deoxynucleoside triphosphates (dNTPs) (Qbiogene, Carlsband, CA, USA), 4 μL reverse transcription (RT) buffer (Promega, Fitchburg, WI, USA), 0.5 U Moloney murine leukemia virus (M-MLV) reverse transcriptase RNase H Minus, point mutant (Promega, Fitchburg, WI, USA) in a total volume of 20 μL. mRNA samples were also prepared without reverse transcriptase as a control for DNA contamination. After reverse transcription, cDNA samples were quantified using the BioDrop μLITE and subsequently properly diluted in DNAse/RNAse-free water (Promega, Fitchburg, WI, USA) to reach a final concentration of 5 ng/μL.

### 2.5. DNA Extraction

DNA was extracted from bacteria using the InstaGene Matrix Kit (Bio-Rad Laboratories, Hercules, CA, USA) and the final concentration was determined by measuring the absorbance at 260 nm and the 260/280 nm ratio using the BioDrop μLITE (BioDrop, Milan, Italy).

### 2.6. Genes Selection and Primer Tests on *Lactobacillus paracasei* A13 

The enzymes investigated derived from the pathways of Membrane FA metabolism and the UFA synthesis. Sequences of the primary enzymes involved in these pathways were identified through the on-line platform KEGG PATHWAY Database (http://www.genome.jp/kegg/pathway.html) using different strains of *Lb. paracasei* (*Lb. paracasei* 8700:2, *Lb. paracasei* ATCC 334, *Lb. paracasei* N 1115, *Lb. casei* BL 23, *Lb. casei* LC2W, *Lb. reuteri* DSM 20016). Then, the sequences were aligned using the CLUSTALΩ software (http://www.ebi.ac.uk/Tools/msa/clustalo/) in order to define the conservative and non-conservative regions. Based on that, Primer3web version 4.0.0 software (http://primer3.ut.ee/) was used to design the primers. Each primer was validated with Beacon Designer software (http://www.premierbiosoft.com/qOligo/Oligo.jsp?PID=1). For each primer pair, the amplification specificity and the optimal conditions for the qPCR reaction were assessed using the genomic DNA of *Lb. paracasei* A13, as a template. Different MgCl_2_ concentrations (2.0, 3.0, 4.0 mM) and annealing temperatures (AT) were tested (60, 62 and 64 °C). The quality of amplification was verified by gel electrophoresis using 1.5% (*w*/*v*) agarose gels (data not showed). 

### 2.7. Real-Time PCR (RT-qPCR) 

RT-qPCRs were performed using a Rotor gene 6000 thermal cycler (Corbett Life Science, Mortlake, Australia), according to Braschi et al. [21]. Briefly, the RT-PCR reaction mixture (2.5 μL) included 5 ng of cDNA, 6.5 μL of SYBR Premix Ex Taq II (TaKaRa Bio Inc., Shiga, Japan), 0.25 μM of each primer and 4.5 μL of DNAse/RNAse free water (Promega, Wisconsin, USA). Using the Rotor-Gene series software (Qiagen Inc., Ontario, Canada), the threshold line and quantitative cycle (Cq) of each gene were determined. The list of genes, their function and optimal RT-qPCR conditions (MgCl_2_ concentration and annealing temperature_)_ are reported in Table 1.

### 2.8. Relative Gene Expression and Statistical Analysis

Relative Gene Expression (RGE) was determined according to MIQE guidelines [23] using the mathematical model proposed by Pfaffl [24] and reviewed by Muller et al. [25]. Based on the literature, Reference Genes (RGs) *ileS* (Isoleucine tRNA ligase), *lepA* (Elongation factor 4), *pyrG* (CTP synthase) and *pcrA* (ATP-dependent DNA helicase) were chosen [22], analyzed in relation to their contents in Guanine-Cytosine and evaluated using different statistical parameters by the BestKeeper © tool program [26,27]. Three independent replicates were performed for all experiments. The RGEs detected for each gene in relation to the homogenization pressure applied were considered significant (*p* < 0.05) on the basis of ANOVA and TUKEY HSD post-hoc test performed using the 0.1 MPa treatment as the control condition. Data were processed using STATISTICA 8 (Version 8.0; Statsoft., Tulsa, OK, USA).

## 3. Results

Cell loads of *Lb. paracasei* A13 before and after different HPH treatments (0.1, 50, 150 and 200 MPa) are reported in Table 2. Results confirmed the resistance of the strain to the HPH, when compared to the pre-treatment and the control (treated at environmental pressure, 0.1 MPa). In fact, independently of the severity of the treatment applied, no significant difference was observed in the final cell loads, which ranged between 9.0 and 9.2 log CFU/mL.

Interestingly, the different HPH treatments gave rise to characteristic FA profiles, as shown in Table 3. Modifications were more noticeable in samples treated with 150 and 200 MPa than 50 MPa, where the FA profile was more similar to the control one.

The main FAs detected in *Lb. paracasei* A13, independently to the hyperbaric treatment, were C12:0, 2-OH C12:0, 3-OH C12:0, C13:0, C14:0, C15:0, 2-OH C14:0, 3-OH C14:0, C16 iso, C16:1 cis 9, C16:0, C18:1 9cis, C18:1 9trans, C18:0 and C19 cyc. In general, HPH treated cells were characterized by increased concentrations of both saturated fatty acids (SFAs), such as C12:0, C13:0, C14:0, C16:0 and C18:0, and UFAs such as C16:1, C18:1 cis and C18:1 trans. These increases were more pronounced for UFAs than for the saturated ones. In fact, although both SFA and UFA concentrations increased in almost all HPH treated cells, the ratio UFA/SFA reached its maximum score at 150 MPa. Moreover, the fatty acid chain length (CL) resulted in higher for the cells treated at 150 and 200 MPa. The cells treated at 50 MPa showed a concentration of total FAs similar to the control cells and lower compared to the other treated cell samples. Eventually, a progressive increase of pressure severity led to an increased concentration of cyclic fatty acids (CFAs), such as C19cyc, and a decrease of 2 and 3 hydroxy acids with 12 and 13 carbon atoms. 

To assess the direct effect of sub-lethal HPH treatments on the expression of key genes involved in the fatty acid biosynthesis of *Lb. paracasei* (Figure 1), RT-qPCRs were performed. The genes were taken into consideration and their description are listed in Table 1.

The gene expression data were normalized using *IleS* as the reference gene. This gene was chosen on the basis of the pairwise correlation values in the conditions adopted according to the method of Plaff [26] (data not showed). Specifically, the stability order in our experimental condition of the tested RGs was *ileS* > *lepA* > *pyrG* > *pcrA*. 

Relative gene expression data reported in Figure 2 showed clearly that sublethal HPH treatments, mainly at 150 and 200 MPa, induced an up-regulation of several genes involved in FA biosynthesis pathway. 

The application of a hyperbaric treatment at 50 MPa did not induce significant differences in the expression of genes involved in the FA biosynthesis pathway compared to the control samples with the exception of a significant (*p < 0.05*) up-regulation of *fabH* gene. The samples treated at the highest homogenization pressures (150 and 200 MPa) showed significant (*p < 0.05*) up-regulation of the genes *accA*, *accC*, *fabD*, *fabH* and *fabZ* compared to the controls. In addition, the treatment at 200 MPa induced a significant (*p < 0.05*) up-regulation of *fabK* and *fabZ* compared to the other treated or control samples.

## 4. Discussion

*Lb. paracasei* A13 is an interesting bacterial strain able to produce high-quality functional cheeses and fermented milks. It has been already used to elucidate the mechanisms of action of HPH sub-lethal treatments [18,28,29]. Moreover, its functional performances as probiotic have been widely reported, both in in vitro and in vivo (murine model) models, and they have been described as enhanced by sub-lethal HPH treatments [16,19,30]. However, the mechanisms of adaptation of this strain to mild HPH treatments, especially in terms of cell membrane changes, and the genes involved in the stress response, are not completely understood. In this study genes related to the membrane fatty acid biosynthesis were selected since the modulation of the membrane composition and fluidity is considered by a wide literature a universal microbial adaptation mechanism [14,17,31,32,33,34,35,36,37,38,39]. 

The data obtained in this work showed that the cell load of *Lb. paracasei* A13, when subjected to the different pressure levels adopted, remained almost unchanged. This confirms the high barotolerance of the strain, as reported in the literature, even at 200 MPa [17,19,40]. On the other hand, the capability to survive to a physical treatment is very important for probiotic microorganisms since it is considered one of the most important traits among the selection criteria for their application in foods and beverages [41,42].

As expected, HPH treatments influenced the membrane FA composition of *Lb. paracasei* A13. Although during the HPH treatment an increase of temperature was detected due to the dissipation of a fraction of the mechanical energy as heat in the fluid [9,43], such temperature increase did not result in the appearance of heat damage indicators, such as lactoperoxidase and alkaline phosphatase activities in HPH treated milk or loss of vitamin C in fruit juices [44,45]. The lack of such indicators is attributable to a very short timespan of HPH treatment. In fact, each fluid fraction is treated at the established pressure levels (50–200 MPa) for milliseconds fractions [46,47]. Moreover, in our case, the temperature increase was immediately reduced to 15°C by the application of an external exchanger. On the other hand, the lack of heat damages due to HPH treatments was observed in several food matrices subjected to HPH treatments also at levels higher than 200 MPa for 2 and 3 repeated cycles [8]. If the instantaneous temperature increase was not able to induce the production of heat damage indicators in the suspension media or the food matrices, it surely affected the response mechanisms of the treated cells of *Lb. paracasei* A13. In fact, according to a wide literature [48,49,50], stressful conditions, including those determined by HPH treatments, result in an oxidative stress for microbial cells, due to the imbalance and disproportion between cellular anabolism and catabolism. In such conditions, microbial cells can accumulate reactive oxygen species if adequate response mechanisms responsible for their removal are not activated by the cells. The cell oxidative damage can be even worse applying HPH treatments. In fact, during the passage through the homogenizing valve, an increase of temperature, even if of millisecond fraction, takes place together with the rise in gas partition into cytoplasmic membranes [7,51]. Therefore, the increase in temperature and in oxygen partition result in a marked oxidative stress for the cells that can be counteracted more or less effectively, depending on microbial species and strain and conditions of treatments. However, HPH treated cells, especially the ones subjected to 150 and 200 MPa, were characterized by the significant increase of UFAs. The increase of unsaturation level and specific unsaturated FAs is recognized as the key mechanism to counteract the oxidative damages induced by physico-chemical stressors, including HPH, to microbial cells ([51,52]). It was observed in many lactic acid bacteria, including *Lb. helveticus* and *Lb. paracasei A13*, and yeasts belonging to the genus *Saccharomyces* and *Yarrowia* [14,16,17,19,53,54]. Since desaturases are oxygen-dependent enzymes, the introduction of double bonds inside the carbon chains is seen as a mechanism used by microbial cells to reduce oxidative damages induced by the stressful conditions applied [48,54]. The importance of specific UFAs, such as oleic and linoleic acids, as a general stress response mechanism in gram-positive and gram-negative bacteria and yeasts was recognized by several authors [34,36,39,55]. The hyperbaric treatments induced also a significant increase of C18:1 trans isomer. Although the data on the effect of trans isomers on membrane fluidity are conflicting, the key role in the prevention of the entrance in the microbial cell of toxic molecules is widely recognized [32,33,34,35,36,56]. 

Since the fluidizing mechanisms of UFAs is well known, the increase of FA length was relevant in the *Lb. paracasei* A13 cells after HPH treatment. In fact, the modulation of FA length, highlighted in this work, represents a common response mechanism adopted by different microorganisms to survive in adverse conditions, for example, at low pH and temperatures or during exposure to antimicrobial compounds contributing to the maintenance of the proper fluidity (and consequently the proper functionality) of the cell membrane [48]. Further, our data showed an increase of CFAs upon HPH treatments. These FAs are synthesized from C18:1 Δ9 and C18:1 Δ11. Although their production has been already described and occurs in LAB as a stress response mechanism, their role in the cytoplasmic membrane fluidity has not yet been completely clarified [17,31,57,58]. Our data seem to confirm their role, together with long-chain FAs, in reducing the fluidizing effect of the marked increase of UFAs. In addition, HPH treatment induced a decrease in the concentrations of the 2 and 3 hydroxy acids, having 12 and 13 carbon atoms, when the pressure level increased. Hydroxy acids, together with the 6-carbon aldehydes and the lactones, are part of the so-called oxylipins which are synthesized, starting from polyunsaturated fatty acids, by several lactic acid bacteria under stress conditions [59,60,61]. Their concentration decreases with the rise of the severity of the hyperbaric treatment can be explained with the well-known ability of many lactic bacteria to transform these molecules, when exposed to osmotic, heat and oxidative stress conditions, into *quorum sensing* molecules such as furanone [59,62,63]. 

The phenotypic modulation in membrane fatty acid composition induced by HPH treatments in *Lb. paracasei* A13 cells was confirmed by the gene expression data. In fact, a significant perturbation in the expression of *Lb. paracasei* A13 key genes involved in fatty acid biosynthesis were also observed in response to HPH treatment. As the pressure level increased, the over-expression of *fabD* and *fabH* was observed. These genes are involved in FA initiation and elongation, supporting the increase of long-chain FAs observed in our experimental conditions. *fabD* catalyses the first step of initiation, transferring the malonyl group of malonyl-CoA to the acyl carrier protein (ACP) and forming malonyl-ACP, the elongation substrate for fatty acid biosynthesis. On the other hand, the major condensation reactions in the initiation of type II FA biosynthesis is catalyzed by *fabH* [64,65]. Therefore, their significant up-regulation after hyperbaric treatments at 150 and 200 MPa justifies the increase also in CFAs recorded. Overexpression of *fabH* was reported by Fernandez et al. [66] as an adaptive mechanism adopted by *Lb. bulgaricus* to counteract acidic stress. Srisukchayakul et al. [67] also showed an up-regulation of *fabH* and *cfa* by 6-fold and 12-fold, respectively, in *Lb. plantarum* NCIMB 8826 under acidic stress conditions. Moreover, it seems that the β-ketoacyl-ACP synthase III, the product of the *fabH* gene, plays a key role in branched-chain fatty acids production by gram-positive organisms, such as bacilli, staphylococci, and streptomycetes [68]. 

Our experimental results showed, after a 200 MPa HPH treatment, a significant upregulation also of *fabK*, the main gene associated with the increase of long-chain FAs in bacterial fatty acids biosynthesis. In fact, the enoyl-acyl carrier protein (enoyl-ACP) reductases (ENRs) expressed by this gene can reduce trans-2-enoyl-ACPs to saturated ACP species [69]. This increase in gene expression after HPH treatments could be another strategy adopted by *Lb. paracasei* A13 to counteract the pronounced increase in the unsaturation and to maintain proper cell membrane fluidity and functionality. Other genes that increased their expression by HPH treatments were *fabZ*, particularly at 150 and 200 MPa, and *fabG* and *fabF*, but only when 200 MPa were applied. *fabZ* codifies for a dehydratase that can introduce double bonds inside the carbon chain, playing a key role in unsaturated FA production [70]. The hyperexpression of the desaturase genes under stress conditions associated to the increase of UFAs in microbial cell membranes is outlined by a wide literature concerning different microbial species and strains and different sublethal growth conditions [48,53,55]. For example, the increase of *fabZ* and *fabG* expression associated with a hyperproduction of UFAs was observed by Beld et al. [71] in *E. coli* strains exposed to stressful conditions. Moreover, high expression of *fabZ* was associated with a significant increase in specific UFAs such as cis-C16:1 and cis-C18:1 [66]. *fabF*, instead, is involved in a two-carbon atom elongation of the FA chain starting from C16:1. In *E. coli*, this gene also plays a crucial role in palmitoleate elongation at low temperatures, producing cis-vaccinate [72]. Siroli et al. [33], showed an overexpression of the whole *fab* gene family when *E. coli* K12 was exposed to different natural antimicrobials. The up-regulation of these genes was also related to an increase in unsaturated, branched and cyclic FAs biosynthesis.

Our results showed that *Lb. paracasei* A13 activated a series of reactions aimed to control and maintain the membrane fluidity, and consequently its functionality, in response to HPH treatments. In fact, the fluidizing effect of the UFA synthesis seems in our experimental conditions to be counteracted by the treated cells turning a part of them to CFAs and elongating the length of carbon chains of straight and branched FAs. However, it has to be emphasized that bacterial FA biosynthesis is a very complex phenomenon that has to be quickly regulated in stressful conditions in order to preserve the viability and the activities of the microorganisms. For instance, Serrazanetti et al. [14] showed that *S. bayanus* cells exposed to HPH treatment at 80 MPa increased the level of unsaturation together with an increased expression of OLE1, ERG3 and ERG11 immediately after HPH treatment. In fact, they were upregulated in the first 5 min after the treatment. These genes are involved in UFAs and sterols biosynthesis, and they are fundamental for yeast membranes to modulate their fluidity and to maintain their functionality, under stress conditions. The up- and down-regulations of genes involved in FA biosynthesis have been reported for several bacteria including *Lb. plantarum*, *Bifidobacterium longum* [73,74] and *E. coli* [33]. 

## 5. Conclusions

The results obtained in this work represent an important step towards understanding the response mechanisms of *Lb. paracasei* A13 when subjected to sub-lethal HPH treatments, both in terms of membrane fatty acid modulation and response of genes involved in fatty acid biosynthesis. This study indicates that *Lb. paracasei* A13 cell membrane counteracts the sub-lethal stress caused by hyperbaric treatments through modulations of lipid composition. In fact, an increase in membrane unsaturation level, counterbalanced by a rise of CFAs, was observed. The gene expression data proved that the regulation of the expression of *fab* genes represents an immediate response mechanism adopted by *Lb. paracasei* A13 to HPH treatments aimed to balance the membrane fluidity through the regulation of genes involved in fatty acids biosynthesis. Although further studies aimed to understand the modulations of genes involved in membrane phospholipids and glycoproteins biosynthesis will be necessary to understand the mechanisms regulating the interaction with host cells, this study represent a first step in the comprehension of the adopted mechanisms by *Lb. paracasei* A13 in order to better exploit its functionality in food formulation.

## Figures and Tables

**Figure 1 microorganisms-08-00439-f001:**
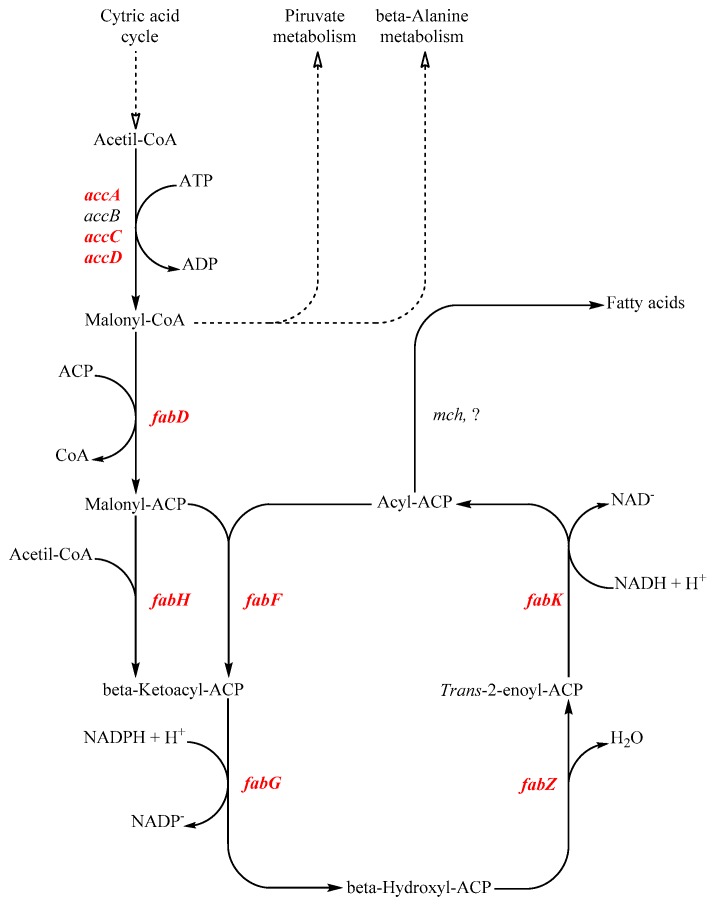
Schematic representation of the conserved fatty acids biosynthesis pathway based on the *Lb. paracasei* ATCC334 fatty acids synthesis model. In red are highlighted the genes considered in this study. Acc genes (*accA*, *accB*, *accC*, *accD*) are involved in the malonyl-CoA synthesis. Malonyl-ACP Malonyl-CoA:ACP transacylase (*fabD*) catalyze the coenzyme A substitution with an acyl carrier protein (ACP). Malonyl-ACP is condensed by 3-oxoacyl-ACP synthase (*fabH*) to acetyl-CoA, producing a β-ketoacyl-ACP. Ketoacyl-ACP reductase (*fabG*) starts with a reduction and is followed by a dehydration executed by 3-hydroxyacyl-ACP- dehydratase (*fabZ*). A second reduction by enoyl-ACP-reductase (*fabK*) produces acyl-ACP. In order to continue the elongation reaction, acyl-ACPs are dehydrated by 3-oxoacyl-ACP synthase (*fabF*) and the elongation chain cycle starts again. Once sufficiently elongated, acyl-ACPs are processed by different hydrolytic enzymes depending on the chain length. Medium-chain fatty acids (i.e., C:12) are released by the esterase activity of the Medium-chain acyl-ACP hydrolase (*mch*).

**Figure 2 microorganisms-08-00439-f002:**
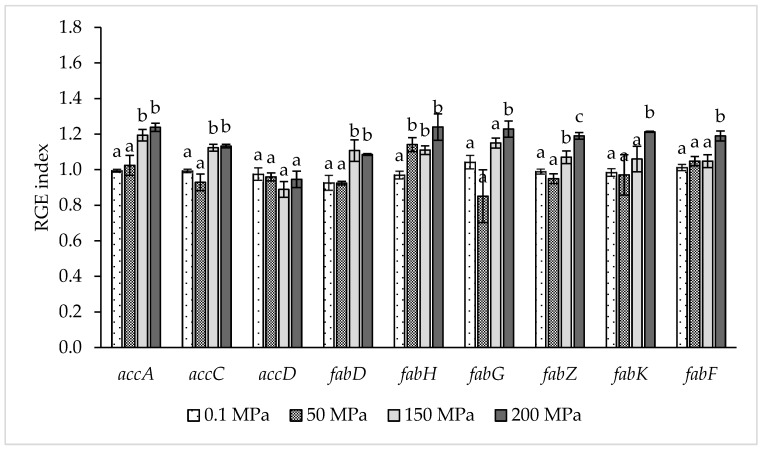
Relative gene expression (RGE) index of *Lb. paracasei* fatty acids biosynthesis genes accA, accB, *accD*, *fabD*, *fabH*, *fabG*, *fabZ*, *fabK* and *fabF* in relation to homogenization pressure. RGE index was calculated according to the model proposed by Pfaffl [24] and reviewed by Muller et al. [25] using *ileS* (Isoleucine tRNA ligase) as reference gene and homogenization at 0.1 MPa as the control condition. The results are the average of three replicates and for the same gene, column followed by different superscript letters are significantly different *p* < 0.05. Error bars indicate ± SD.

**Table 1 microorganisms-08-00439-t001:** Primers used in this work and their description in terms of function, annealing temperature, final concentrations of MgCl_2_ and relative reference. *ileS*, *lepA*, *pyrG*, *pcrA* were tested as reference genes for the gene expression trials.

Gene	Primer Sequence (5′ > 3′)	Biological Function	MgCl_2_ (mM)	Annealing Temperature (°C)	Reference
*accA*	F: CCGGCGCATATCCTGGCAAA	acetyl-CoA carboxylase carboxyl transferase subunit alpha	2	62	This study
	R: ACCTCATCCCCAAACGCCAA				
*accC*	F: ACTTCACCTCGCTGCGACCA	acetyl-CoA carboxylase, biotin carboxylase subunit	2	64	This study
	R: TGCCACTTCATCGCCAGCTG				
*accD*	F: TTGCCCGCACTGCCATTACG	acetyl-CoA carboxylase carboxyl transferase subunit beta	3	62	This study
	R: TCGCCTGACCTGTCCACACA				
*fabD*	F: GCATTGCCCGATGAAACCGC	malonyl CoA-acyl carrier protein transacylase	2	62	This study
	R: GCACTTGTTGACAGGCAGCC				
*fabH*	F: TTTGCTGATGGTGCTGGCGG	3-oxoacyl-ACP-synthase III	4	64	This study
	R: ACCGCCCGCCCATTCATTTT				
*fabG*	F: GCGTGCGAGTCAATGCGATC	3-oxoacyl-ACP-reductase	2	62	This study
	R: ACCAGAAAACGGGCCGCATG				
*fabZ*	F: CGATGCCGGACTTCAAAGGA	(3R)-hydroxymyristoyl-ACP-dehydratase	2	62	This study
	R: ACTAAATCCGCACTGCTGGC				
*fabK*	F: TTGCTGACGGTCGTGGTGTG	enoyl-ACP-protein reductase	3	62	This study
	R: CGGTCGTGTCAGAAAGGGCA				
*fabF*	F: TTTACTGGGTGCGGCTGGTG	3-oxoacyl-ACP-synthase II	2	62	This study
	R: ATGCCCGCCGAAACCAAAGG				
*mch*	F: CGGTTTTGCGGGTGATCGGT	acyl-ACP thioesterase	3	62	This study
	R: TGGTTCCTCAATTCGCGGCA				
*ileS (RG)*	F: ACCATTCCGGCTAACTATGG	Isoleucine tRNA ligase	2	64	Alcántara et al. [22]
	R: TCAGGATCTTCGGATTTTCC				
*lepA (RG)*	F: CACATTGATCACGGGAAGTC	Elongation factor 4	2	64	Alcántara et al. [22]
	R: GTAATGCCACGTTCACGTTC				
*pyrG (RG)*	F: AATTGCGCTTTTCACTGATG	CTP synthase	2	64	Alcántara et al. [22]
	R: CGAAATGATCGACCACAATC				
*pcrA (RG)*	F: CGGCCAATAATGTGATTCAG	ATP-dependent DNA helicase	2	64	Alcántara et al. [22]
	R: TCATCAGTTTCGCTTTGAGC				

**Table 2 microorganisms-08-00439-t002:** Viable colony-forming units ± Standard Deviation (SD) of *Lb. paracasei* A13 before and after HPH treatment at 0.1, 50, 150 and 200 MPa. Means followed by different letters are significantly different *p < 0.05.*

Pressure MPa	log CFU/mL ± SD
Pre-Treatment	9.18 ± 0.11 ^a^
0.1	9.20 ± 0.09 ^a^
50	9.00 ± 0.15 ^a^
150	9.16 ± 0.08 ^a^
200	9.20 ± 0.12 ^a^

**Table 3 microorganisms-08-00439-t003:** Membrane FA composition of *Lb. paracasei* A13 in relation to the HPH treatments applied. Values are expressed as ppm ± SD. In the same line relative to each fatty acid, means ± SD followed by different superscript letters (a, b, c or d) are significantly different *p* < 0.05. Missing values (-) were under the detection limit.

	MPa			
Fatty Acid	0.1 (Control)	50	150	200
C12:0	100.0 ± 3.4 ^a^	120.5 ± 3.2 ^b^	180.1 ± 5.4 ^c^	181.7 ± 6.2 ^c^
2-OH C12:0	1.8 ± 0.1	-	-	-
C13:0	2.9 ± 0.1 ^a^	11.8 ± 0.4 ^b^	61.1 ± 2.5 ^d^	46.5 ± 1.6 ^c^
3-OH C12:0	3.4 ± 0.1 ^a^	3.3 ± 0.2 ^a^	-	-
C14:0	18.2 ± 0.5 ^a^	18.5 ± 0.9 ^a^	38.9 ± 1.9 ^b^	38.3 ± 2.1 ^b^
C15:0	10.0 ± 0.2	-	-	-
2-OH C14:0	22.5 ± 1.1 ^a^	11.0 ± 0.8 ^b^	22.4 ± 1.8 ^a^	-
3-OH C14:0	9.1 ± 0.6 ^a^	4.2 ± 0.2 ^b^	-	-
i-16:0	17.3 ± 0.8 ^a^	11.9 ± 0.9 ^b^	-	14.0 ± 0.9 ^b^
C16:1 9 cis	30.5 ± 1.8 ^a^	35.1 ± 1.5 ^a^	59.3 ± 2.3 ^b^	57.8 ± 2.2 ^b^
C16:0	114.6 ± 6.3 ^a^	101.7 ± 5.2 ^a^	145.8 ± 5.2 ^b^	147.0 ± 7.9 ^b^
C18:1 9 cis	150.4 ± 7.5 ^a^	170.2 ± 8.9 ^a^	339.4 ± 18.9 ^c^	280.6 ± 15.3 ^b^
C18:1 9 trans	14.54 ± 1.1 ^a^	19.3 ± 1.9 ^b^	33.1 ± 2.5 ^d^	26.2 ± 1.6 ^c^
C18:0	-	28.1 ± 4.3 ^b^	56.5 ± 4.6 ^c^	34.3 ± 2.3 ^d^
C19Cyc9+C19cyc11	152.1 ± 6.8 ^a^	155.3 ± 7.5 ^a^	292.9 ± 11.5 ^b^	307.5 ± 18.6 ^b^
UFA ^1^	195.4 ± 12.4 ^a^	224.5 ± 13.3 ^b^	431.8 ± 22.6 ^d^	364.6 ± 24.6 ^c^
SFA ^2^	451.9 ± 19.8 ^a^	466.2 ± 15.4 ^a^	797.6 ± 39.8 ^b^	769.3 ± 45.7 ^b^
UFA/SFA ^3^	0.43 ± 0.2	0.48 ± 0.1	0.54 ± 0.3	0.47 ± 0.3
CL ^4^	16.1 ± 0.1	16.4 ± 0.2	16.6 ± 0.2	16.8 ± 0.1

^1^ Unsaturated Fatty Acids; ^2^ Saturated Fatty Acids; ^3^ Unsaturated Fatty Acids/Saturated Fatty Acids ratio; ^4^ Chain Length.

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
