# Peer review of "Lactobacillus paracasei A13 and High-Pressure Homogenization Stress Response"

_microorganisms, 2020, doi:10.3390/microorganisms8030439_

Round 1

Reviewer 1 Report

The authors have made some corrections and have improved the overall quality of the manuscript.

Author Response

Referee 1

The overall quality of the manuscript was improved and the manuscript revised according to. Also English was improved.

Reviewer 2 Report

In the revised manuscript, titled “Lactobacillus paracasei A13 and high pressure homogenization stress response,” Siroli et al. demonstrated the changes in the levels of fatty acids and transcripts for select genes after high pressure homogenization (HPH) treatment. While no additional experiments were performed in the revised manuscript, the authors have addressed this reviewer’s previous comments with the following remaining concerns.
Major
1. This reviewer thinks the possibility of HPH treatment increasing the efficiency of lipid extraction or RNA extraction should be considered. While the colony forming units (CFUs) show there is no lethal effects of the HPH treatment, CFUs represent both intact as well as damaged but recovered bacterial cells after HPH treatment. With such a drastic increase in UFAs and SFAs, a more efficient extraction seems a logical explanation. Since the authors are reluctant to perform any additional experiments (such as killed controls), the authors should at least acknowledge the possibility. It might also help if the authors discussed and compared the degrees of increase in other studies (references 48, 67-69) to support their claim. Moreover, the claim of FA biosynthesis can be “quickly regulated” (line 359) need additional evidence. Description of other studies did not include relevant treatment periods. The millisecond treatment from this study followed by the minute-long cooling needs to be emphasized too.
2. With qRT-PCR, you are measuring the transcript levels, which might be caused by changes in expression or by transcript stability. This is more apparent with the new information on the different stability of reference genes. “Up-regulation” is an interpretation. “Increase in transcript levels” is the measured outcome. The authors might want to consider changing the narratives to match the experimental measurements. Also, because transcript levels for known stress response genes or genes that may alter fatty acid composition (mch) were never measured, the authors should at least acknowledge the lack of positive controls for comparison and propose future approaches in the discussion.
3. There remains numerous grammatical and editorial mistakes that need to be addressed. The authors prefer writing in fragments instead of fully narrative sentences. Paragraphs starting on lines 233, 272, and 282 need to be indented. Also, the use of “some” should also be avoided for better clarity (examples: lines 58, 259 …). What is this “RNase H-point mutant” that was used in reverse transcription?
Minor
1. It seems that the document provided to address the reviewer’s comments was not the final version. A comment box was left toward the end.
2. The authors should consider including “data not shown” in places where they didn’t include the method development results. With the new primers developed for this study, were the products confirmed by sequencing?
3. The cooling duration of 1 minute was not in the revised manuscript as indicated.
4. The aerobic cultivation requires specifications. Were the cultures agitated? If yes, how fast (rpm)? For these organisms, if no agitation was included, the cultures will become anaerobic.
5. Wherever percentages were reported, either v/v or g/v should be indicated for clarity.
6. This reviewer is not sure what the authors referred to as “heat indicators” (line 275). Maybe plain language for all areas of microbiology will be helpful.
7. In Table 3, can the UFA/SFA ratio and chain length be calculated for each sample so standard deviation can be included as well?
8. Clarifications are needed for lines 248-251. It appears that the two sentences are describing opposing results.

Author Response

Major
1. This reviewer thinks the possibility of HPH treatment increasing the efficiency of lipid extraction or RNA extraction should be considered. While the colony forming units (CFUs) show there is no lethal effects of the HPH treatment, CFUs represent both intact as well as damaged but recovered bacterial cells after HPH treatment. With such a drastic increase in UFAs and SFAs, a more efficient extraction seems a logical explanation. Since the authors are reluctant to perform any additional experiments (such as killed controls), the authors should at least acknowledge the possibility. It might also help if the authors discussed and compared the degrees of increase in other studies (references 48, 67-69) to support their claim. Moreover, the claim of FA biosynthesis can be “quickly regulated” (line 359) need additional evidence. Description of other studies did not include relevant treatment periods. The millisecond treatment from this study followed by the minute-long cooling needs to be emphasized too.

We would like to thank the Referee for the careful revision of our manuscript. We have taken into consideration all the criticisms and the suggestions and we have revised according to them. Probably some criticisms were due to lack of clearness of some sentences. Now we have revised and re-written the discussion in order to clarify our point of view better discussing it in accordance with the literature. The time of HPH treatment has been précised and the difference between its effects on the food matrix and cell suspension medium clarified according to the literature available. In fact, the time of treatment is of few milliseconds. Such span time generally assure a reduction of thermal damage in food systems compared to traditional heat treatments. By contrast, it is reported to increase the oxidative stress of microbial cells similarly to many physic-chemical stresses. Dodd in 1997 was the first to demonstrate that all the stressful conditions result in microbial cells in the unbalance between anabolism and catabolism with consequent increase of oxidative stress. A wide literature relative to a wide range of stresses (including HPH) and microorganisms is in agreement with Dodd et al. (1997) data.

We have discussed better than in the previous version these aspects and additional references have been introduced. Also, the sentence on the quickly regulation claim of FA biosynthesis has been clarified and other references has been introduced.

  1. With qRT-PCR, you are measuring the transcript levels, which might be caused by changes in expression or by transcript stability. This is more apparent with the new information on the different stability of reference genes. “Up-regulation” is an interpretation. “Increase in transcript levels” is the measured outcome. The authors might want to consider changing the narratives to match the experimental measurements. Also, because transcript levels for known stress response genes or genes that may alter fatty acid composition (mch) were never measured, the authors should at least acknowledge the lack of positive controls for comparison and propose future approaches in the discussion.

We agree with the Review on the importance of the reference genes. In fact, we consider such gene and also other 3 other reference genes. We normalized data using only IleS as reference gene because the stability order in our experimental condition of the tested RGs was ileS>lepA>pyrG>pcrA. This procedure is reported in a wide literature. We have also done several repetitions and replicates in order to increase the significance of our data.

  1. There remains numerous grammatical and editorial mistakes that need to be addressed. The authors prefer writing in fragments instead of fully narrative sentences. Paragraphs starting on lines 233, 272, and 282 need to be indented. Also, the use of “some” should also be avoided for better clarity (examples: lines 58, 259 …). What is this “RNase H-point mutant” that was used in reverse transcription?

We revised the manuscript following the reviewer suggestions

“RNase H-point mutant” is referred to the Promega product “M-MLV reverse transcriptase RNase H Minus, point mutant” (Promega). We revised the sentence in material and method section. We used this product because, as reported in manufacturer instruction, Moloney Murine Leukemia Virus Reverse Transcriptase, RNase H Minus (M-MLV RT [H–]), Point Mutant, is an RNA-dependent DNA polymerase that can be used in cDNA synthesis with long RNA templates (>5kb). The lack of RNase H activity is beneficial for this application, as RNase H can start to degrade templates when incubation times are long, as they may be when synthesizing long cDNAs.

Minor
1. It seems that the document provided to address the reviewer’s comments was not the final version. A comment box was left toward the end.

We are sorry for the file version; we will upload the correct final version this time.
2. The authors should consider including “data not shown” in places where they didn’t include the method development results. With the new primers developed for this study, were the products confirmed by sequencing?

The primers were specifically designed based on the nucleotides sequence of specific genes using Primer3web and each primer was validated using specific in vitro software tools (Beacon Designer software). In addition, For each primer pair the amplification specificity was verified by end point PCR using Lb. paracasei A13 genomic DNA as template. If only one DNA band was present on 1.5% agarose gel for each primer pair, we assumed that the amplification was relative to the gene for which the primers were built.

We improved the text in the new version of the revised manuscript to better explain this point.

  1. The cooling duration of 1 minute was not in the revised manuscript as indicated.

We included the information in the new version of the revised manuscript
4. The aerobic cultivation requires specifications. Were the cultures agitated? If yes, how fast (rpm)? For these organisms, if no agitation was included, the cultures will become anaerobic.

The control cells and the treated ones were grown in the same conditions to standardize the effects of the cultivation conditions on the modulation of membrane fatty acids and gene expression. However, we revised the manuscript according to the review criticisms. In fact, we specify in material and method section that we have not used agitation during the culture grow but we left an abundant headspace.

  1. Wherever percentages were reported, either v/v or g/v should be indicated for clarity.

We reported v/v or w/v to percentage reported throughout the manuscript
6. This reviewer is not sure what the authors referred to as “heat indicators” (line 275). Maybe plain language for all areas of microbiology will be helpful.

We revised the sentence as follow “heat damage indicator such as lactoperoxidase and alkaline phosphatase activity in milk or loss of vitamin C in fruit juices
7. In Table 3, can the UFA/SFA ratio and chain length be calculated for each sample so standard deviation can be included as well?

The standard deviation has been included for UFA/SFA and CL in table 3
8. Clarifications are needed for lines 248-251. It appears that the two sentences are describing opposing results.

The sentences reported in lines 248-251 have been revised as follow: “The application of a hyperbaric treatment at 50 MPa did not induce significant differences in the expression of genes involved in the FA biosynthesis pathway compared to the control samples with the exception of a significant (p<0.05) up-regulation of fabH gene”

This manuscript is a resubmission of an earlier submission. The following is a list of the peer review reports and author responses from that submission.

Round 1

Reviewer 1 Report

The manuscript investigates a very important and topical issue in contemporary food microbiology and molecular biology: the effect of HPH on a probiotic strain of lactic acid bacteria and the molecular and molecular-genetic mechanisms of the strain to withstand to the sub-lethal stress. The manuscript is well-organized, detailed, yet concise; the methods are described correctly and in detail; the results are well-presented, interpreted and discussed; the conclusions are logically drawn from the results and their discussion; the practical application of the results obtained is clearly stated in the end of the conclusion. I have no notes and recommendations. Thus, my decision is accept the manuscript for publication as is.

Reviewer 2 Report

In the article, titled “High pressure homogenization and Lactobacillus paracasei A13: stress response and challenge to increase its functionality,” authors Siroli et al. reported the fatty acid composition and relative gene expressions of L. paracasei after high pressure homogenization treatment. As an important probiotic and food bacterium, L. paracasei must survive and maintain its functions after antimicrobial treatments. Therefore, better understanding of how the organism responds to antimicrobial treatment processes is of relevance to food safety. However, this manuscript provides very preliminary results that require additional experiments to reach a more relevant conclusion. Therefore, this reviewer does not believe this manuscript is suitable for publication at the moment.

Major Concerns

The entire introduction section was written in a single paragraph. This reviewer highly recommends organizing the related ideas into separate paragraphs. One paragraph should focus exclusively on HPH, including the physical mechanisms relevant to food preservation. One paragraph should focus on lactic acid bacteria and their functions post-treatment. One paragraph should focus on sharing more information on the model organism paracasei A13, including what is known and what is not known. The rationale for the reported experiments needs to be much better stated. Additionally, considering this journal is not catered only to food microbiologists, the authors should take care in describing terminology used in food microbiology. For example, it is not clear what “food diversification” means (line 38). Many of the sentences also include numerous qualifying clauses and lists. The authors should consider rephrasing toward a more narrative style. Descriptions in the Materials and Methods lack sufficient details. How was paracasei A13 grown? There was mentioning of anaerobic conditions but how the anaerobic conditions were established needs to be clarified. If the model organism was a commercial strain, the vendor source should be included. How long did it take for cultures to cool down to 10°C in ice water? The section 2.3 feels redundant. The timing of sample collections for different assays should be described with the assay, not as a separate section. What is the purpose of 1 hour versus 24 hours of recovery? Where were these cultures used? Section 2.7 title should be shortened. It is also not clear how the reference genes were being used. Four genes were listed as reference genes but only one was used to calculate relative expression. If the annealing temperatures were different for different genes, were the samples run separately at different temperatures or at one temperature? Finally, the authors should avoid citing references for additional descriptions of methods. At least briefly describe the key pieces of information, if not the details of the entire procedure. Were the observations, in fatty acid composition or relative gene expression, based on pressure changes or temperature changes? With 2.5°C increase for ever 10 MPa, the higher-pressure samples were also exposed to a much higher temperature. Since the “cooling down” part likely takes longer than the milliseconds of HPH treatment, the authors should also consider the effects of the cool down stage. What is known about paracasei A13 temperature responses? The overall increase in total fatty acids requires additional interpretation. How long does it usually take for bacteria to change fatty acid composition? Typically, studies report fatty acids as percentage to better illustrate the changes in the overall composition. Can the HPH treatment/cooling alter the subsequent lipid extraction process? The rationale behind selecting the genes tested in this study was not clearly articulated. Are there already known genes that up- or down-regulated under high pressure or high temperature? The authors mentioned the generation of oxidative stress. Are there genes that have been shown to be involved in directly eliminating ROS? “Overexpression” is a term used to describe conditions where a gene was upregulated on purpose using a controlled system. With barely 1.2 fold increase, the gene expression results were definitely not an example of “overexpression” and were modest at best. With the limited number genes being analyzed, these results were hardly “transcriptomic” data (line 25).

Minor Concerns

For data reported in Table 2, instead of referring the number as “cell load,” the authors might consider using “viable colony forming units.” Because of the incubation period required for colonies to develop, the number of colonies represent viable bacteria (damaged or not damaged) present at the end of HPH treatment. As the title suggests, this study aims to study the “functionality” of paracasei. However, no subsequent functionality studies were performed after the HPH treatment. A few simple functional demonstrations, such as lactic acid production or LDH activity assay might help support the claim. The manuscript definitely requires a more thorough editing for typos and grammatical errors. Some paragraphs need to be separated while others need to be combined.